Prognosis and immune landscape of bladder cancer can be predicted using a novel miRNA signature associated with cuproptosis

Zhang Zhilei 1
Liu Fang 2
Yu Yongbo 1
Xie Fei 3
Zhu Tao 1 zhutao0922@126.com
1 The Department of Urology, The First Affiliated Hospital of Shandong Second Medical University , Weifang , China
2 The Department of Cardiology, The First Affiliated Hospital of Shandong Second Medical University , Weifang , China
3 The Department of Urology, The Affiliated Hospital of Qingdao University , Qingdao , China
Jurisic Vladimir
Electronic publication date: 2024 Nov 29
Publication date: 2024
Volume: 12
Electronic Location ID: e18530
Received 2024 May 30; Accepted 2024 Oct 24
Copyright: © 2024 Zhang et al.
Copyright year: 2024
Copyright holder: Zhang et al.
License: This is an open access article distributed under the terms of the Creative Commons Attribution License, which permits unrestricted use, distribution, reproduction and adaptation in any medium and for any purpose provided that it is properly attributed. For attribution, the original author(s), title, publication source (PeerJ) and either DOI or URL of the article must be cited.
License URL: https://creativecommons.org/licenses/by/4.0/

Keywords: Cuproptosis, miRNA, Prognosis, Immune, Drug sensitivity, Bladder cancer

Funding: Project ZR2020QH238 (Shandong Provincial Natural Science Foundation) This work was supported by Project ZR2020QH238 (Shandong Provincial Natural Science Foundation). The funders had no role in study design, data collection and analysis, decision to publish, or preparation of the manuscript.

==============================
Background

Bladder cancer is characterized by a high recurrence rate and mortality, posing a significant challenge to clinical management. Recently, cuproptosis, a novel form of regulated cell death, has been identified as a potential target for therapeutic intervention in various diseases. The contribution of cuproptosis-related microRNAs (miRNAs) in bladder cancer pathogenesis, however, remains largely unexplored. Therefore, the current study aims to construct a miRNA signature related to cuproptosis for predicting the prognosis and facilitating personalized therapeutic strategies in bladder cancer patients.

Methods

In this study, we retrieved transcriptomic data and clinical information pertaining to bladder cancer from publicly available databases, including the Cancer Genome Atlas (TCGA) and Gene Expression Omnibus (GEO). We identified a set of 19 cuproptosis-related genes through a comprehensive review of relevant literature. Using multivariate Cox regression and LASSO analysis, we constructed a cuproptosis-related miRNA prognostic signature. The Kaplan-Meier (K-M) and receiver operating characteristic (ROC) curves were used to validate the accuracy of prediction. Additionally, we developed a nomogram incorporating clinical characteristics and the miRNA signature to further assess its prognostic value. We evaluated the tumor microenvironment (TME) of every patient using immune ESTIMATE, CIBERSORT, and ssGSEA algorithms. We also investigated the differences in tumor mutation burden (TMB) and drug sensitivity between two groups. Finally, we validated the prognostic value of this miRNA signature using the OncomiR dataset.

Results

We developed a panel of eight cuproptosis-associated miRNAs to serve as a prognostic signature. The predictive validity of this signature was determined using Kaplan-Meier and ROC curves, and was found to be acceptable in both the TCGA training, test and total dataset. The prognostic value of this signature was confirmed by univariate and multivariate Cox regression analysis, indicating its applicability as a prognostic factor. The immune cell infiltration was significantly associated with an immunosuppressive phenotype of TME in the high-risk group of patients; meanwhile, patients in the high-risk group had a lower TMB resulted in shorter survival. Notably, higher estimate scores and IC50 value for chemotherapy drugs were observed in the high-risk group, indicating poor response to immune therapy and chemotherapy.

Introduction

Bladder cancer, the second most common malignant tumor in males, and globally ranks as the sixth most commonly diagnosed malignant tumor in men (Sung et al., 2021; Siegel, Miller & statistics, 2020). Most bladder tumors arise from the epithelium lining of the urinary tract. Among them, urothelial bladder cancer (UBC) accounts for over 90% of cases and is further classified into non-muscle invasive bladder cancer comprising Tis, Ta, and T1 stages, and muscle-invasive bladder cancer (MIBC) ranging from T2 to T4 (Sanli et al., 2017; Prasad, Decastro & Steinberg, 2011). MIBC is present in approximately 20–25% of bladder cancer patients and is associated with a poor 5-years survival rate, even with aggressive treatments such as surgery, radiotherapy and chemotherapy (Moschini et al., 2016).

Although disease-specific mortality has declined recently due to significant advances in disease diagnosis and treatment, the prognosis of treatment response and personalized post-treatment management still requires improvement (Antoni et al., 2017). The high degree of heterogeneity in bladder cancer poses a challenge for predicting clinical outcomes. Consequently, there is a pressing need to develop a new prognostic model that enable personalized assessment of cancer risk. The management of bladder cancer involves lifelong monitoring strategies, which include frequent and invasive cystoscopy, imposing a negative impact on the quality of life of patients and incurring substantial financial costs on the healthcare system (Bhanvadia, 2018). Identification of novel molecular markers can enhance prognosis and risk stratification, ultimately reducing the need for unnecessary intervention.

Recently, a study published in Science has identified a novel form of copper-induced cell death, different from apoptosis, necrosis, ferroptosis and pyroptosis (Tsvetkov et al., 2022). The mechanism involves direct binding of copper to lipoylated mitochondrial enzymes, leading to protein toxicity stress and instability of iron-sulfur (Fe-S) cluster proteins, named “cuproptosis”, and several genes have been associated with this process (Tsvetkov et al., 2019). There is mounting evidence suggesting that copper levels may impact occurrence of cancer (Bian, Fan & Xie, 2022). Moreover, previous investigators found cuproptosis plays a role in several types of cancers, including renal cell carcinoma (Xu et al., 2022), hepatocellular carcinoma (Zhang, Sun & Zhang, 2022), and melanoma (Lv et al., 2022), with cuproptosis gene signatures utilized for prediction of the prognosis and immune response. While several studies have explored cuproptosis-related gene signature for predicting the prognosis of bladder cancer patients (Zhang et al., 2022a; Bai et al., 2022; Zhang et al., 2022b). To the best of our knowledge, no studies about cuproptosis-related miRNA in bladder cancer was reported. In this study, a novel cuproptosis-related miRNA signature was constructed to predict immune landscape and drug sensitivity of the patient with different cuproptosis scores in BLCA.

Materials and Methods

Data collection

RNA sequencing data and clinical characterization for BLCA were retrieved from the TCGA database as of November 28th, 2022. The dataset comprised 418 tumor samples and 19 normal tissue samples, and RNA-Seq data of miRNAs were also obtained according to the gene annotations from TCGA. Additionally, one Gene Expression Omnibus (GEO) dataset, GSE13507, consisting of 256 samples, was obtained from GEO.

Cell lines

The human bladder urothelial carcinoma cell line T24 and UMUC3 and human normal urine epithelial cell line SV-HUC-1 were supplied by the Cell Bank of the Chinese Academy of Sciences. The cell lines were cultivated in standard DMEM medium (MedChem Express, Monmouth Junction, NJ, USA) supplemented with 10% foetal calf serum (FCS) and 1% penicillin‒streptomycin (P/S) purchased from Gibco (Grand Island, NY, USA).

Tissue samples

Paired human samples of BLCA tissues and neighbouring ordinary mucosa were obtained from patients treated at the First Affiliated Hospital of Shandong Second Medical University (n = 3) for the following FISH assay. All cases were diagnosed as primary BLCA without any treatment before surgery. Informed consent was obtained from all participants before sample collection. This research received permission (No. KYLL20240613-1) from the ethical committee of the First Affiliated Hospital of Shandong Second Medical University and was conducted according to the Helsinki.

Screening and differential expression analysis of cuproptosis-associated miRNAs

Firstly, we conducted a co-expression correlation analysis of miRNA and cuproptosis-related gene expression profiles using the “limma” “edgeR” and “pheatmap” packages. We obtained 19 cuproptosis related genes from previous studies (Tsvetkov et al., 2022; Xu et al., 2022; Zhang, Sun & Zhang, 2022). miRNAs differentially expressed between normal and bladder cancer tissues were identified using a Log2FC cutoff of > 1.5 and p < 0.01.

Construction of the prognostic cuproptosis related miRNA signature

The BLAC samples were divided into training and testing groups randomly. Univariate Cox regression was conducted in the training group, and cuproptosis related miRNAs were identified by Lasso. These miRNAs were further analyzed by multivariate Cox regression analysis, and only those with a p value < 0.01 were selected as the basis for the best model for signature construction. Then, risk scores were investigated using the equation: Risk score = Exp miRNA1 × β miRNA1 + Exp miRNA2 × β miRNA2 + Exp miRNA3 × β miRNA3 + • + Exp miRNAn × β miRNAn.

Nomogram and calibration

To develop a comprehensive prognostic model for bladder cancer patients, the risk scores were combined with various clinical pathological factors, and a nomogram was constructed. The nomogram was designed to predict overall survival (OS) rates of patients. The Hosmer-Lemeshow test was applied to draw calibration curves, which assessed the predictive performance of the nomogram.

PCA, GO, and KEGG analysis

miRNAs associated with cuproptosis in bladder cancer samples were analyzed using PCA analysis to visualize distribution of samples. The intersected genes from different risk groups and different immune-score groups were then subjected to Gene Ontology (GO) analysis, which comprised three components: biological process (BP), cellular component (CC), and molecular function (MF). Additionally, differentially expressed KEGG pathways were evaluated using the Hs. “eg.db”, “clusterProfiler”, and “enrichplot” packages.

Immune profile analysis

Immune cells and mediators in oncology are directly involved in the process of cell death in the tumor microenvironment (TME) (Hinshaw & Shevde, 2019). To analyze the relationship of immunity and stromal in TME, the immune and stromal score of each sample was calculated by the “estimate” package. We also utilized the CIBERSORT algorithm to evaluate infiltration of 22 immune cells in TME, which including Macrophages M0, Macrophages M1, Macrophages M2, T cells CD4 naive, T cells CD4 memory resting, T cells CD4 memory activated, T cells follicular helper, T cells regulatory (Tregs), NK cells resting, NK cells activated, T cells gamma delta, neutrophils, monocyters, CD8+ T cells, plasma cells, mast cells resting, mast cells activated, dendritic cells resting, dendritic cells activated, B cells naive, B cells memory.

TMB analysis and pharmaceutical screening

The mutation data was retrieved from the TCGA website, and then integrated and analyzed using the “maftools” package. Differences in TMB and survival rates were assessed between high-risk and low-risk groups, with a significance threshold of <0.05. Furthermore, the IC50 values of drugs commonly employed in bladder cancer were predicted for the high- and low-risk groups using “pRRophetic” package.

Validation the prognostic of cuproptosis‑related miRNAs in OncomiR datasets

Bladder cancer patients were stratified into high and low expression groups based on the levels of cuproptosis-related miRNA expression in the OncomiR online database. Additionally, a signature composed of eight cuproptosis-related miRNAs was established and applied to assess the survival analysis of these miRNAs in the OncomiR dataset.

Quantitative real-time polymerase chain reaction

We first extracted total RNA and reverse transcribed into cDNA using a reverse transcription kit (Accurate Biotechnology (Human) Co., Ltd., Guangzhou, China), according to the manufacturer’s instructions. All steps were performed according to the manufacturer’s instructions and our previous study (Yu et al., 2024). Primers for amplification were synthesized by Huada Gene (Beijing, China) and were listed in Table S1. U6 was used as the internal control of miRNA.

Fluorescence in situ hybridization

The subcellular localization of four kinds of miRNA were measured using the fluorescence in situ hybridization (FISH) kit (F11101/50 and F21101/50, Genepharma, shanghai, China). The cell and tissue slides were treated with miRNA probe hybridization solution labeled by SA-Cy3 according to the manufacturer’s instructions. Brifely, the slide was hybridized at 42 °C for 16 h and immersed in 2 × SSC (saline sodium citrate buffer), followed by immersion in 70% ethanol for 3 min and stained with DAPI for 10 min. The slide was imaged using the Olympus fluorescence microscope and AxioVision image analysis software. Probes for FISH were listed in Table S2.

Transwell migration and invasion assays

All steps were performed as our previous study (Yu et al., 2024). Brifely, the BLCA cells (5 × 104 cells/well) were placed on the upper Transwell chamber (8 μm, Corning, NY, USA) with complete culture medium incubated for 24 h, then stained with 0.1% crystal violet after fixed with 10% formaldehyde. Images were taken under a light microscope. The sequence of interfered inhibitors and mimics for cells were listed in Tables S3 and S4.

Statistics

The quantitative variables were analyzed by the independent-samples t-test. Univariate and multivariate Cox regression analysis were performed to identfy the independent prognostic factors of OS. The nomogram was constructed using the survival and rms package in R based on the coefficients of the multivariable model. The Kaplan-Meier method was utilized to estimate the survival rate of OS between groups, and the log-rank test was employed to compare significant differences. The discriminatory performance of the nomogram was assessed by calculating AUC. Calibration curves were also plotted to compare the predicted nomogram results with the actual Kaplan–Meier estimates of OS possibility in the cohort. The significant level in the present study was two tailed p < 0.05.

Results

The expression of cuproptosis related genes in bladder cancer and prognosis-related miRNAs with coexpression of cuproptosis

As depicted in Fig. 1A, approximately nine out of nineteen cuproptosis-related genes, including NFE2L2, NLRP3, ATP7A, SLC31A1, LIPT2, MTF1, CDKN2A, GCSH and DLST, displayed significant differences between normal and bladder cancer tissues in the TCGA database. In addition, these cuproptosis related genes were all significantly associated with the prognosis of bladder cancer patients in both TCGA and GEO dataset (Figs. 1B–1F, Figs. S1A–S1G). Subsequently, the TargetScan database was utilized to screen 25,702 miRNAs as cuproptosis related miRNAs. In TCGA database, 135 up-regulated miRNAs and 96 down-regulated miRNAs were identified as differentially expressed DE-miRNAs (Figs. 1G, 1H).

Figure 1 Expression levels of cuproptosis related genes and prognosis-related miRNAs with co-expression of cuproptosis in bladder cancer.

(A) Expression levels of nineteen genes in normal bladder tissues and BLCA. (B–G) A prognostic value analysis of ATP7B, CDKN2A, LIAS, LIPT1, NFE2L2, DLST. (H, I) The heatmap and volcano Plot DE-miRNAs in TCGA-BLCA database. (J, K) miRNAs identified by the LASSO-Cox regression model in TCGA-BLCA cohort. *p < 0.05, **p < 0.01, ***p < 0.001.

Construction of cuproptosis-related miRNA signature related to prognosis of bladder cancer patients in TCGA cohort

A total of 61 candidate miRNA with a p-value less than 0.01 were identified by univariate cox regression analysis. Subsequently, these miRNAs were further screened using LASSO-Cox regression analysis (Figs. 1I, 1J). Ultimately, a miRNA signature consisting of eight-miRNA was developed through multivariate cox regression analysis. The corresponding risk scores of bladder cancer sample were calculated using multivariate cox regression analysis in the TCGA datasets, where the risk score was computed as follows: miR-125b-2-3p * 0.43647 − let-7c-5p * 0.45759 + miR-145-3p * 0.32586 + miR-409-3p * 0.22515 + miR-548a-5p * 1.69183 − miR-5682 * 0.49332 − miR-625-3p * 0.27289 + miR-634 * 30.88948. All of the eight miRNAs were significantly associated with patients’ prognosis (Fig. 2A). The overall survival rate of patients in the high-risk group was significantly shorter than that of those in the low-risk group in overall, training and testing groups (Fig. 2B).

Figure 2 Prognostic significance of the eight miRNAs and risk model.

(A) Prognostic value analysis of the screened eight miRNAs from the TCGA-BLCA cohort. (B) Analysis of survival outcomes between high- and low-risk subgroups in training, testing, and TCGA cohorts.

Survival analysis of the signature and independent analysis of prognostic factors

Risk curve suggested that the mortality rate of high-risk patients was higher than that of low-risk patients (Fig. 3A). The heatmap demonstrated that the eight miRNAs could be used to discriminate between high- and low risk levels, where let-7c-5p, miR-125b-2-3p and miR-548a-5p were the high-risk miRNAs, whereas miR-625-3p was the low-risk miRNAs (Fig. 3B). Additionally, the predictive accuracy of the risk score was evaluated using ROC curves for the all, training and testing group, with the AUC of the three groups for the risk score being 0.711, 0.736 and 0.687 respectively (Fig. 3C).

Figure 3 Survival analysis of the signature and construction of a nomogram.

(A) Risk score distribution and relationship between risk score and survival time in the training, testing, and TCGA cohorts. (B) Heatmap of the expression of eight miRNAs in the high- and low-risk subgroups. (C) ROC analysis of the risk score in the three cohorts. (D) Multivariate independent prognostic analysis to analyze that the Cuproptosis-associated miRNAs signature was shown to be an independent risk factor for patient. (E) Nomogram combined with risk score and clinicopathological features (*p < 0.05, **p < 0.01, ***p < 0.001). (F) Calibration curves for 1, 3, and 5 years.

Furthermore, univariate and multivariate cox regression analysis were employed to determine whether the risk score might be independent of other clinical features as a prognostic factor. The results of multivariate cox regression analysis indicated that age (HR = 1.021, 1.004–1.04, P < 0.05), gender (HR = 0.654, 0.460–0.929; P < 0.05), T (HR = 1.844, 1.223–2.779; P < 0.05) and risk score (HR = 3.424, 2.342–5.004; P < 0.05) were significantly associated with overall survival, indicating that the risk signature is an independent prognostic factor for patients with bladder cancer (Fig. 3D). A nomogram including independent prognostic factors was constructed to evaluate the prognosis of bladder cancer patients at 1, 3, and 5 years (Fig. 3E). Moreover, the calibration curve of nomogram model for the prognosis of survival probability displayed a high consistency between the prognostic and measured values (Fig. 3F).

Stratified prognostic analyses of patient clinical features

Patients with bladder cancer were stratified based on various clinicopathological characteristics and molecular subtype to investigate the association between survival probability and risk score. Notably, the overall survival rate was markedly higher in the low-risk group than in the high-risk group across most subgroups, except for the low-grade group (Figs. 4A–4H). These findings imply that the risk score model has promising prognostic value for predicting the survival outcome of bladder cancer patients with diverse clinicopathological features.

Figure 4 Stratified prognostic analyses of patient clinical Features by this risk signature.

(A–H) K-plan–Meier survival curves with stratified age, gender, grade, molecular subtypes, stages, T stage, N stage, M stage in the TCGA-BLCA cohort.

Immune characteristics

To investigate the significance of miRNA risk status in tumor microenvironment, we quantified the level of immune cell infiltration between high-risk and low-risk groups of bladder cancer using a heatmap (Fig. 5A). The results suggested that the CD8+ T cells and CD4+ T cells were negatively correlated with the risk score, while Treg cells, CD4+ memory T cells, mast cells and M0 macrophages were positively associated with the risk score (Figs. 5B–5G). The correlations of infiltrating immune cells and related functions were presented in (Figs. 5H, 5I). Specifically, CD8+ T cells, DCs cells and Th1 cells were positively correlated with tumor-infiltrating lymphocytes (TIL) (Fig. 5H). In addition, a positive association was found between T-cell co-inhibition and check point in the immune function (Fig. 5I). We also used the TIDE score to evaluate whether the BLCA patients would respond well to ICB therapy. Our findings indicated that the high-risk group had a higher TIDE score, although there was no significant difference (Fig. S1H). Furthermore, with regard to TME scores, stromal scores and ESTIMATE scores were higher in high-risk patients, but there was no difference in immune scores between the two risk groups (Fig. 5J). Overall, these results suggest that miRNA risk status could be used as biomarker for predicting the infiltration of immune cells and TME status in bladder cancer.

Figure 5 Immune-related analyses.

(A) Heatmap of infiltrating immune cells related with these miRNAs and risk signature. (B–G) The correlation between risk score and immune cell infiltration analyzed by CYBERSORT. (H, I) Correlation of tumor-infiltrating immune cells and related pathways. (J) The TME score analysis of samples between the high-risk and low-risk groups. *p < 0.05, **p < 0.01, ***p < 0.001.

The principal component analysis and biological pathways analyses

Principal component analysis (PCA) was employed to examine the distinctions between two groups in terms of four expression profiles (cuproptosis associated miRNAs, total gene expression profiles, cuproptosis genes, and risk model) (Figs. 6A–6D). Risk model exhibited the greatest discriminatory capability in distinguishing the low and high-risk populations. The Venn diagram in Fig. 6E depicts the cross-genes among different risk scores and different immune scores (Fig. 6E). KEGG analysis of these genes indicated a predominance of focal adhesion and PI3K-AKT signaling pathway (Fig. 6F), while Gene Ontology analysis revealed a strong association between cuproptosis associated genes and extracellular adhesion (Fig. 6G). Furthermore, a correlation analysis between the intersection genes and immune cells following univariate cox regression indicated a positive correlation between these genes and Treg cells and M0 macrophage, and a negative correlation with CD8+ T cells (Fig. 6H).

Figure 6 The principal component analysis and functional enrichment analysis.

(A) Cuproptosis-related miRNA module. (B) Total gene module. (C) Cuproptosis-related gene module. (D) Cuproptosis-related miRNA prognostic module. (E) The venn diagram of 669 overlapping genes between the risk score and immune score. (F, G) GO and KEGG Enrichment analysis of these overlapping genes. (H) Immune-related functions of genes by univariate cox regression analysis.

TMB analysis and drug sensitivity analysis

We examined mutations in cohorts of both high-risk and low-risk individuals. The results indicated that for the majority of genes, the mutation frequency in the low-risk group was higher than that in the high-risk group (TP53: low risk, 49%; high risk, 48%. TTN: low risk, 49%; high risk, 38%; MUC16: low risk, 26%; high risk, 25%) (Figs. 7A, 7B). Additionally, a statistically significant difference in TMB was observed between the high-risk and low-risk cohorts (Fig. 7C). Moreover, survival analysis suggested a probable distinction between patients with high and low TMB, with those in the high TMB group exhibiting significantly better overall survival than those in the low TMB group (Figs. 7D, 7E). The present study utilized the “pRRophetic” package to identify potentially effective antitumor drugs for bladder cancer treatment, including gemcitabine, 5-fluorouracil, methotrexate, mitomycin, doxorubicin, and axitinib. Subsequently, the sensitivity of these drugs was evaluated, revealing that the high-risk group displayed a higher IC50 value (concentration that inhibits cell growth by 50%) compared to the low-risk group for all drugs except axitinib, suggesting that high-risk patients may have lower sensitivity to these antitumor drugs (Figs. 8A–8F).

Figure 7 The relationship between TMB and the signature.

(A, B) The mutation frequency of the top 15 mutation genes in bladder cancer for the high-risk (216 samples) and low-risk (187 samples) groups were showed by Waterfall plot. (C) Differential TMB between the high-risk group and low-risk group in bladder cancer. (D, E) Survival curves among different groups of patients with bladder cancer.

Figure 8 The IC50 of drug analysis.

(A–E) Boxplots showed the IC50 value of the conventional chemotherapy drug in bladder cancer treatment between the two risk groups. (F) Boxplot demonstrated the IC50 value of the targeted drug in bladder cancer treatment in the high and low risk groups.

Prognostic analysis of the external database and validation of cuproptosis-related miRNAs in vitro experimental

The current study examined a set of eight cuproptosis related miRNAs (hsa-let-7c-5p, hsa-miR-125b-2-3p, hsa-miR-145-3p, hsa-miR-409-3p, hsa-miR-548a-5p, hsa-miR-625-3p, hsa-miR-634 and hsa-miR-5682) listed in OncomiR database. With the exception of hsa-miR-5682, all eight miRNAs were detected in the observed sample (Figs. 9A–9D and Figs. S2A–S2C). Further, survival analysis were conducted to evaluate the association between single cuproptosis-related miRNAs and prognosis of patients with bladder cancer, with results indicating that four of the seven miRNAs analyzed were significantly associated with patient prognosis (Figs. 9A–9D). In addition, a miRNAs signature constructed to evaluate the prognosis of bladder cancer patients suggested that patients with high risk score demonstrated a poor prognosis than the patients with low risk score (Fig. 9E). These findings were consistent with the results obtained from the TCGA dataset. Moreover, to further evaluate the prognostic value of this cuproptosis associated miRNAs, in vitro cell experiments were performed to assess the expression level of the four identified miRNAs. As the RT-qPCR results suggested, these four miRNAs had an overall trend of increased expression levels in bladder cancer cells compared to bladder normal cells (Figs. 9F–9I). In addition, the FISH experiment confirmed the expression levels of four miRNAs in bladder cancer cell lines T24 and UMUC-3. It was found that miR-125-b-2-3p, miR-145-3p, Mir-4093p, and miR-625-3p were expressed in both cell lines (Figs. 10A, 10B). Additionally, these four miRNAs showed significantly higher expression in three pairs of bladder cancer tissues compared to adjacent tissues (Figs. 10C, 10D). Functional verification revealed that all four miRNAs promoted bladder cancer migration, invasion (Figs. 10E, 10F), and proliferation (Figs. 10G–10J) to varying degrees.

Figure 9 Verification of the prognosis of the four miRNAs in the external OncomiR dataset.

(A–D) Prognostic value analysis of four miRNAs in the OncomiR dataset. (E) The survival analysis between high- and low-risk groups divided by the miRNA signature in the OncomiR cohorts. (F–I) RT-qPCR was used to examine the expression of these four cuproptosis-related miRNA in bladder cancer cells and bladder normal cell. *p < 0.05, **p < 0.01, ***p < 0.001, ****p < 0.0001.

Figure 10 Detection and functional validation of four miRNAs in BLCA cells and tissues.

(A, B) The expression levels of four miRNAs were verified by FISH assay in bladder cancer cell lines T24 and UMUC-3. Scare bar = 100 μm. (C, D) Expression of four miRNAs in paracancer and cancer in tissue samples from bladder cancer patients by FISH assay. Scare bar = 100 μm. (E, F) Differences in migration and invasion ability following overexpression or knockdown of four miRNAs in T24 and UMUC-3. Scare bar = 100 μm. (G–J) Differences in proliferation ability after overexpression or knockdown of four miRNAs in T24 and UMUC-3. *p < 0.05, **p < 0.01, ***p < 0.001, ****p < 0.0001.

Discussion

Bladder cancer is a highly aggressive tumor with a poor survival rate (Grayson, 2017). Despite pathological evaluation and AJCC TNM staging being the main diagnostic and prognostic methods for bladder cancer, they are not accurate or sensitive enough to account for its high tumor heterogeneity. Patients with the same clinical stage often have different therapeutic effects and clinical outcomes, necessitating the exploration of new molecular biomarkers to improve patient prognosis and quality of life. Copper, as an organic metal ion, plays an important role in many physiological processes in the human body, which helps maintain intracellular copper concentration at very low levels through the body’s balance mechanism (Williams, 1983). Copper-induced death occurs primarily in the TCA cycle, where copper combines with acetylated lipids leading to the accumulation of fatty acylated proteins and loss of iron–sulfur cluster proteins, resulting in cell death due to excessive toxic proteins (Chang et al., 2021). Emerging evidence suggests that curoptosis plays a crucial role in tumor cell proliferation, growth, and metastasis (Oliveri, 2022). Copper consumption has been found to reduce angiogenesis and lead to cell death due to its role as an antiangiogenic factor (Chan et al., 2020). Target therapy of curoptosis is being applied to clinical research.

MicroRNAs (miRNAs) are endogenous non-coding RNA with regulatory function found in eukaryotes, about 20 to 25 nucleotides in length. miRNA can inhibit the transcription and translation of target mRNA, but can also shear target mRNA and promote its degradation, indicating its critical role in the development regulation and potential use as potential markers in multiple cancers (Rupaimoole & Slack, 2017; Jet et al., 2021). However, the prognostic signatures of miRNA targeting cuproptosis related genes have not been studied in bladder cancer.

This study constructed a gene signature of eight miRNAs targeting 19 CRGs in bladder cancer tissues, which had good prognostic value for bladder cancer based on the model’s risk score. In addition, the verification of prognostic analysis in stratified group demonstrated that the miRNAs feature had excellent performance in the prognosis. A new nomogram was established by combining the miRNAs risk model with clinical factors to evaluate the prognosis of patients with bladder cancer, demonstrating an accurate prediction of the 1-, 3-, and 5-year survival rates. At the same time, the results of survival curve suggested that the model was suitable for a variety of different clinical pathological features of bladder cancer patients, which could also be applied to the molecular subtypes of luminal and basal.

As usual, different immune cells infiltration are associated with various tumor progressions (Sierra et al., 2021). The dysregulated antitumor immunity in the tumor microenvironment was closely related to tumorigenesis, progression and invasion (O’Brien & Finlay, 2019). CD4+ Treg cells, M2 macrophage and MDSCs were usually related with immune suppression. The high-risk group was positively correlated with this immunosuppressive phenotype by these immune cells infiltration, leading to a poorer prognosis for the high-risk group, while NK cells, CD8+ T cells and their mediators also play an important role in tumor elimination.

Furthermore, ESTIMATE scores, stromal scores and, immune scores were calculated between the two different subgroups of the population, and the higher-risk groups had higher stromal scores and higher tumor purity. As mentioned earlier, the TIDE algorithm is used to evaluate patients’ clinical response to immune checkpoint inhibitors (ICI) treatment; with a higher TIDE score suggesting a greater possibility of immune escape, thus leading to the occasion that ICI treatment for the higher TIDE score patients will not have an effective response.

In this study, although no difference was found in TIDE scores between patients in the high-risk and low-risk groups, patients in the high-risk group may exhibit a more restricted response to ICI therapy for the TME of immunosuppressive phenotype. Moreover, the DEGs were identified using risk score and immune score, and the GO and KEGG analyses revealed that these DEGs were primarily involved in extracellular structure organization and ECM receptor interaction.

TMB, which reflects the capacity and degree of neoantigen production, can predict the effectiveness of immunotherapy for various tumors (Jardim et al., 2021; Fusco, West & Walko, 2021; McGrail et al., 2021). Study reported that copper was closely related with genetic mutations in tumors (Fanni et al., 2021). In this study, a significant difference in TMB was observed between the high-risk and low-risk groups, with the high-risk group exhibiting a lower TMB than the low-risk group. Additionally, bladder cancer patients with high TMB had a better prognosis than those with low TMB, which is consistent with previous studies (Lv et al., 2020). Furthermore, patients with a lower risk score exhibited a better prognosis compared with patients with a lower TMB score, which implied that the risk score of this model can independently predict immunotherapy response and can be used for further study.

In addition, IC50 is an important index to evaluate drug efficacy for treatment response, with higher IC50 values indicating a greater possibility of drug resistance in cancer patients (Geeleher, Cox & Huang, 2014). Next, the pRRophetic algorithm was used to screen effective drugs for chemotherapy and targeted therapy of bladder cancer based on IC50 values. Patients in the high-risk group were less sensitive to these chemotherapy drugs, which can aid doctors in making appropriate treatment decisions for different patients. In bladder cancer, the low-risk group was more inclined to conventional chemotherapy like Gemcitabine, 5-Fluorouracil, Methotrexate, Mitomycin, and Doxorubicin, while the high-risk group was more prone to the targeted drug Axitinib. Futhermore, the screened four miRNAs were verified to be highly expressed in bladder cancer tissues and could promote its progression to varying degrees. Among them, mir-125-b-2-3p and mir-409-3p had stronger promoting effects on bladder cancer. This discovery offers a novel orientation for the clinical treatment of bladder cancer.

Moreover, recent research underscores the significant role of the urinary microbiome in bladder cancer pathogenesis. Nardelli et al. (2023) recently identified specific urinary bacteria, that are linked with bladder cancer, particularly influencing carcinogenesis through mechanisms of chronic inflammation. This emerging evidence complements our findings by suggesting that the urobiome could interact with molecular pathways, including those involving cuproptosis-related miRNAs, potentially affecting the immune landscape and tumor behavior. Further studies could explore how these microbial elements correlate with the miRNA signatures we have identified, potentially opening new avenues for integrated diagnostic and therapeutic strategies.

In addition, beyond molecular and genetic markers, systemic inflammatory responses have also been recognized as significant predictors of oncological outcomes in bladder cancer. For instance, a recent research demonstrated that the systemic immune-inflammation index (SII), derived from preoperative blood tests, is a potent predictor of lymph node invasion, advanced pathological stages, and overall survival in patients undergoing radical cystectomy (Russo et al., 2023). This finding underscores the role of systemic inflammation as an integral part of the tumor microenvironment, which may interact with molecular pathways, including those involving miRNAs. Considering the link between elevated SII and poor prognostic outcomes, integrating systemic inflammatory markers with cuproptosis-related miRNA signatures could offer a more comprehensive approach to predict and manage bladder cancer effectively.

The risk signature was validated in internal and external datasets; however, there were some limitations and deficiencies in this study. Firstly, the lack of our own data to confirm the investigation results necessitates further validation in vitro and in vivo. Secondly, due to the lack of clinical follow-up data, the prognostic value of the model remains to be proven.

Conclusions

In total, a prognostic signature is constructed based on the screened cuproptosis-related miRNAs in bladder cancer patients, which has been validated as a reliable prognostic model. Furthermore, this signature can aid in identifying immune cell infiltration, immune function, tumor microenvironment and drug sensitivity of bladder cancer. Additionally, a nomogram was constructed by combining the risk score signature and clinical factors, demonstrating high predictive accuracy. In conclusion, the gene signature system not only facilitates understanding the tumor characteristic of bladder cancer patients but also aids in improving the prognosis and individualized treatment of bladder cancer.

Supplemental Information

Supplemental Information 1 Supplemental Tables.

Supplemental Information 2 Supplemental Figures.

Supplemental Information 3 Raw data or Figure_9: qRT-PCR.

The Ct level of miRNAs

We acknowledge the GEO database for providing their platforms and contributors for uploading their meaningful datasets. We also thank all participants involved in studies included in our present study.

Additional Information and Declarations

Competing Interests

Author Contributions

Data Availability

The authors declare that they have no competing interests.

Zhilei Zhang performed the experiments, prepared figures and/or tables, and approved the final draft.

Fang Liu analyzed the data, prepared figures and/or tables, and approved the final draft.

Yongbo Yu analyzed the data, prepared figures and/or tables, and approved the final draft.

Fei Xie conceived and designed the experiments, authored or reviewed drafts of the article, and approved the final draft.

Tao Zhu conceived and designed the experiments, authored or reviewed drafts of the article, and approved the final draft.

The following information was supplied regarding data availability:

The RNA sequencing data and clinical characterization for BLCA are available at TCGA: https://portal.gdc.cancer.gov/projects/TCGA-BLCA, November 28th, 2022. The dataset comprised 418 tumor samples and 19 normal tissue samples, and RNA-Seq data of miRNAs were also obtained according to the gene annotations from TCGA.

The gene expression data is available at GEO: GSE13507.

The raw qRT-PCR data are available in the Supplemental File.

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
