# Peer review of "Prognosis and immune landscape of bladder cancer can be predicted using a novel miRNA signature associated with cuproptosis"

_PeerJ, doi:10.7717/peerj.18530_

## Round 0.1 · original submission · Major Revisions

It is necessary for the authors to correct the paper in many aspects according to the instructions of the 3 reviewers

Besides that:

For immune tests, the material and methods section does not describe the cell types at all, as well as the program used to analyze the data and which cells were analyzed. Only in the discussion are certain cell populations mentioned in relation to tumor stroma interactions, such as CD4, while NK cells and their mediators also play an important role in tumor elimination.

It is necessary to describe the research method in more detail, as well as to add references that show the importance of studying immune cells as well as mediators in oncology, which are directly related to cell shedding in tumor stroma interaction

Reviewer 1 ·

Basic reporting

This work is relatively preliminary without solid validaiton.

Experimental design

This is almost a dry lab work. Very limited cell line study had been performed but the significance is limited.

Validity of the findings

There is no solid validation in real tissue cohort performed.

Additional comments

1. The authors should validate their findings in well-established in-house cohort using ISH for targeted miRNAs.
2. The authors should perform functional validation in cell lines to confirm the story in biological aspects.
3. The dry lab works are performed using commercial tools and only a few novelties are identified.

Reviewer 2 ·

Basic reporting

Language Clarity and Professionalism: The manuscript is generally well-written with professional language. However, there are occasional grammatical errors and awkward phrasing (e.g., redundant use of "prognosis" and "predict" in close proximity) that could be streamlined for better readability.

The background provides a comprehensive review of the relevance of cuproptosis in cancer therapy, but it could better highlight gaps that this study specifically aims to fill, particularly the unique aspects of bladder cancer treatment. for example, incorporating insights into the tumor microenvironment's complexity, recent research underscores the significant role of the urinary microbiome in bladder cancer pathogenesis. For instance, identified specific urinary bacteria, that are linked with bladder cancer, particularly influencing carcinogenesis through mechanisms of chronic inflammation (PMID: 38298766). This emerging evidence complements our findings by suggesting that the urobiome could interact with molecular pathways, including those involving cuproptosis-related miRNAs, potentially affecting the immune landscape and tumor behavior. Further studies could explore how these microbial elements correlate with the miRNA signatures we have identified, potentially opening new avenues for integrated diagnostic and therapeutic strategies. please cite it.

Data Transparency: All raw data seem to be appropriately supplied and referenced, which is commendable.

Figure and Table Presentation: Figures and tables are relevant and generally well-presented. However, some figures are overly complex and could benefit from simplification or more detailed legends to aid understanding

Experimental design

Research Question and Scope: The study addresses a clear, relevant, and original research question about the prognostic value of miRNA signatures in bladder cancer linked to cuproptosis.

Validity of the findings

Data Robustness: The data and analytical methods used are robust and appropriate for the research question. The findings are supported by the data provided, with statistical validation conveyed through survival analysis and ROC curves.

Impact and Novelty: The study introduces a novel miRNA signature linked to cuproptosis which could potentially refine prognostic assessments in bladder cancer, a significant step forward. Beyond molecular and genetic markers, systemic inflammatory responses have also been recognized as significant predictors of oncological outcomes in bladder cancer. For instance, a recent research demonstrated that the systemic immune-inflammation index (SII), derived from preoperative blood tests, is a potent predictor of lymph node invasion, advanced pathological stages, and overall survival in patients undergoing radical cystectomy (PMID: 38138166). This finding underscores the role of systemic inflammation as an integral part of the tumor microenvironment, which may interact with molecular pathways, including those involving miRNAs. Considering the link between elevated SII and poor prognostic outcomes, integrating systemic inflammatory markers with cuproptosis-related miRNA signatures could offer a more comprehensive approach to predict and manage bladder cancer effectively. please cite it.

Link Between Conclusions and Original Research Question: The conclusions are directly linked to the original research question and are supported by the results. However, the discussion could better integrate how these findings compare with existing models or predictions in bladder cancer.
Suggested Improvements:

Compare the predictive power of the new miRNA signature against existing prognostic models to highlight its relative performance and benefits.

Additional comments

none

·

Basic reporting

It provides clear and sufficient background context, presented with a professional article structure and accompanied by relevant figures.

Experimental design

The experimental design is well-constructed, relevant, and meaningful. However, the clinical details, including the specifics of drug treatment and response outcomes, may influence survival rates. Please provide the data on clinical characteristics and treatment regimens."

Validity of the findings

The results appear to be meaningful; however, the treatment regimen was found to correlate with survival and should be incorporated into the findings.

Additional comments

These are very interesting research findings, showcasing a well-designed study and a novel discovery. The results should be correlated with clinical data and treatment outcomes for a comprehensive analysis.

---

## Round 0.2 · accepted · Accept

The 2 reviewers suggested accept this paper. Regarding their recommendation my conclusion is accept this paper in current version

Reviewer 1 ·

Basic reporting

The revision is acceptable

Experimental design

The revision is acceptable

Validity of the findings

The revision is acceptable

Additional comments

The revision is acceptable

Reviewer 2 ·

Basic reporting

This article present a well-defined and meaningful research question that fills an identified knowledge gap. I recommend publication.

Experimental design

The investigation was conducted rigorously, upholding high technical and ethical standards. Methods are sufficiently detailed for replication.

Validity of the findings

The conclusions are well-articulated, directly linked to the research question, and supported by the results

Additional comments

non, I recomend publication.